# Forex Investment Optimization Using Instantaneous Stochastic Gradient Ascent—Formulation of an Adaptive Machine Learning Approach

Iqbal Murtza [1,*,†] ![ID], Ayesha Saadia [2], Rabia Basri [1], Azhar Imran [1,†] ![ID], Abdullah Almuhaimeed [3] ![ID] and Abdulkareem Alzahrani [4] ![ID]

1   Department of Creative Technologies, Air University, Islamabad 44000, Pakistan
2   Department of Computer Science, Air University, Islamabad 44000, Pakistan
3   The National Centre for Genomics Technologies and Bioinformatics, King Abdulaziz City for Science and Technology, Riyadh 11442, Saudi Arabia
4   Faculty of Computer Science and Information Technology, Al Baha University, Al Baha 65779, Saudi Arabia
*   Correspondence: iqbal.murtza@mail.au.edu.pk
†   These authors contributed equally to this work.

**Abstract:** In the current complex financial world, paper currencies are vulnerable and unsustainable due to many factors such as current account deficit, gold reserves, dollar reserves, political stability, security, the presence of war in the region, etc. The vulnerabilities not limited to the above, result in fluctuation and instability in the currency values. Considering the devaluation of some Asian countries such as Pakistan, Sri Lanka, Türkiye, and Ukraine, there is a current tendency of some countries to look beyond the SWIFT system. It is not feasible to have reserves in only one currency, and thus, forex markets are likely to have significant growth in their volumes. In this research, we consider this challenge to work on having sustainable forex reserves in multiple world currencies. This research is aimed to overcome their vulnerabilities and, instead, exploit their volatile nature to attain sustainability in forex reserves. In this regard, we work to formulate this problem and propose a forex investment strategy inspired by gradient ascent optimization, a robust iterative optimization algorithm. The dynamic nature of the forex market led us to the formulation and development of the instantaneous stochastic gradient ascent method. Contrary to the conventional gradient ascent optimization, which considers the whole population or its sample, the proposed instantaneous stochastic gradient ascent (ISGA) optimization considers only the next time instance to update the investment strategy. We employed the proposed forex investment strategy on forex data containing one-year multiple currencies' values, and the results are quite profitable as compared to the conventional investment strategies.

**Keywords:** mathematical finance; forex market; machine learning; investment optimization; forex sustainability; forex economy

## 1. Introduction

In this era of transition from paper currency to plastic and digital currencies, the nature of vulnerabilities has gained more complexity. Now, the world currencies are affected by factors not limited to supply and demand, current account deficit, gold reserves, dollar reserves, petroleum prices, political stability, security, and the presence of war in that country or even its neighborhood [1–3]. In this era of the global village, war even in one country creates economic ripples across the world. These factors result in fluctuation in world currencies and, thus, result in instability in their economies [4]. In particular, with the current tendency of looking beyond the existing SWIFT system, the mentioned risk of currency fluctuation is seemingly greater, and this is creating a more complicated environment for world economies, particularly in developing countries [5–9]. The forex

markets (foreign currency exchange markets in which brokers exchange different currencies with a minor difference in the buying and selling rate are called the open forex market; this business activity is also conducted using banking channels, and this is called interbank forex) are are directly affected by such instabilities in local and international economies.

Because of the world's shift to the global village, the volume of forex trading has increased exponentially in recent decades. According to the statistics, the global forex market in 2022 is expected to be about USD 2.409 quadrillion, i.e., approximately USD 6.6 trillion per day for trades in foreign exchange markets [10,11]. Such a huge market volume is directly at potential risk because of global instabilities, and if this market is unstable or unluckily crashed, this will have critical economic effects.

Currently, in forex markets, there is a trend of having a few renowned currencies in reserve for trading. It is noted that the risks of having reserves in a few currencies are more probable as compared to having reserves in all world currencies [12–14]. Unluckily, the latter choice is not feasible for trading, whereas the first choice has related risks. In this regard, having appropriate reserves in multiple currencies, although seeming simple, fortunately, has also the potential to take advantage of the dynamic nature of the global forex market. Although most of the world trade and forex market are dollar-dominant [14,15], the latest trend of countries trading in local currencies is creating the space to have reserves in multiple currencies, thus ensuring more independence in the currency values of other countries or their economies [16–20]. The same problem also exists for individual savings in international currencies. This significance creates the demand for such a study on having reserves in multiple currencies to minimize the related risks and, at the same time, to benefit from the dynamic nature of the forex market. In this research, we consider this challenge to work on the formulation of an automatic and adaptive forex investment strategy by taking advantage of gradient ascent maximization [21,22], applied to the formulated problem.

## 2. Related Work

In the forex exchange market, which is volatile and dynamic in nature [23,24], both conventional and machine learning methods are used to predict the value of financial time series. In the literature, there are both statistical and machine-learning-based approaches available in the domain of financial time series prediction [25].

From statistical techniques that consider forex data as time series, the employment of the autoregressive integrated moving average (ARIMA) model has gained significant attention from the research community [26]. Unluckily, the ARIMA requires stationary data, a property that the forex data may not necessarily have. In this regard, addressing the uncertainty related to the stochastic process to generate the time series, the combination of fuzzy time series with fuzzy ARIMA shows a better forecast than traditional models [27]. In addition to ARIMA and fuzzy ARIMA techniques, the autoregressive moving average (ARMA) [28] and particle-filter-based [29] forecasting are also good candidates among statistical techniques for predicting time series since they do not require stationary data necessarily.

Contrary to these conventional statistical forecasting methods, few researchers use machine learning techniques for financial time series prediction. Machine learning models have already shown promising results for various research domains [30,31]. Among machine learning techniques, there is related work from both domains of conventional and deep learning techniques [32,33]. From conventional machine learning techniques, Kim et al. [34] explored various support vector machines (SVMs) for financial time series forecasting. To maximize SVMs' performance, Jaiwang et al. [35] worked on upgrading the SVM training model using computational intelligence and evolutionary-computing-based optimization techniques. In their model, forecasting the SVM together with genetic algorithm (GA) outperformed other optimization techniques.

Contrary to the conventional machine learning techniques, deep learning techniques have also gained attention in financial time series prediction [32,33]. Among these deep learning techniques, the employment of the multilayer perceptron (MLP), recurrent neural

network (RNN), gated recurrent unit (GRU), and long short-term memory (LSTM) in time series forecasting is given in the literature [33]. The nature of MLP is basically nonlinear regression, which is unable to exploit the past and present data's coexistence relationship [36]. Still, it has been successfully used by Jarusek et al. [37] to exploit Elliot wave patterns for forex rate prediction. Contrary to the conventional neural network, LSTM is capable of exploring and exploiting present and past data relationships, and thus, it outperforms the conventional neural network [36]. In addition to this, LSTM also solves the vanishing gradient problem, resulting in efficient training as compared to the conventional neural networks. Moreover, in this line of research, Saud et al. [38] explored the gated recurrent unit for optimizing the pricing in the Nepal stock exchange market, whereas its employment in the forex market is missing.

Unluckily, all these statistical, machine-learning-, and deep-learning-based classification techniques are static (one-time training) in their nature. Because of this static nature, these techniques are not capable of adaptive learning and, thereby, cannot accommodate the volatile and dynamic nature in the future. Secondly, the available techniques mostly focus on predicting or forecasting the next values, whereas exploiting the next values is not well addressed in the literature. Moreover, based on these forex market predictions, the formulation and development of an optimized investment strategy in this volatile and dynamic economic environment are also not well addressed in the available research. In this regard, we consider this challenge to work on minimizing the losses from the dynamic and volatile nature and, instead, exploit them to maximize sustainability using the stochastic gradient ascent method. This led us to the formulation and the development of a unique flavor of the gradient ascent method, i.e., the instantaneous stochastic gradient ascent method, which surprisingly outperforms conventional investment strategies.

## 3. The Proposed Methodology

Before presenting the formulation of the proposed methodology, were note that we considered the following assumptions for the formulation.

### 3.1. Assumptions

1.  We considered five entities ($n$ = 5) dollar (USD), Euro (EUR), Dirham (AED), rubble (RUB), and gold for the investment strategy;
2.  We considered that the rates of these entities are updated once a day;
3.  The rates of all entities were considered for an open market;
4.  The rates of all entities were considered in Pakistani rupees (PKR);
5.  Fractional buying and purchasing of entities are possible.

These assumptions were made by realizing the forex market trends of Asian countries, i.e., to have renowned currencies in their forex markets. Currently, we considered 5 currencies, and thus, a few renowned currencies may be missing in these 5 currencies, which can be added to the proposed model easily since its formulation is generic, i.e., n currencies. The second and third assumptions were made to standardize the time updating interval as interbank forex rates' update per day in Pakistan. The fourth assumption was made just as an example: any host currency can be considered since the formulation of the proposed model is generic in this regard. The last assumption was made since the formulation of the proposed model may result in a recommendation to buy fractional (for example, 4.51) units of a currency. This assumption was made to simplify the formulation of the proposed model.

### 3.1.1. Problem Statement

Consider that, on the $i$th day, we have $u_{i1}, u_{i2}, \ldots, u_{in}$ units of $n$ currencies worth $x_{i1}, x_{i2}, \ldots, x_{in}$ Pakistani rupees (PKR) according to the rates $r_{i1}, r_{i2}, \ldots, r_{in}$, respectively. Considering the rate updates each day, how should we update the units of each currency to maximize their worth in PKR.

### 3.1.2. Problem Formulation

Consider n currencies to invest and the investment vector for the $i$th day is $\mathbf{x}_i = [x_{i1}, x_{i2}, \ldots, x_{in}]^T$, where $x_{ij}$ is the investment allocated on the $i$th day for the $j$th currency. The total investment $x_i$ on the $i$th day is the Manhattan distance or $l_1$-norm of the investment vector $\mathbf{x}_i$ as follows:

$$x_i = |\mathbf{x}_i|_1 = \sum_{r=1}^{n} x_{ij} \tag{1}$$

Consider on that $i$th day the rates for $n$ currencies are $r_{i1}, r_{i2}, \ldots, r_{in}$, and thus, the rate vector of this day is $\mathbf{r}_i = [r_{i1}, r_{i2}, \ldots, r_{in}]^T$. Thus, the unit buying vector for the $i$th day can be computed from the point-by-point division of the investment vector by the rate vector, i.e., $\mathbf{u}_i = \mathbf{x}_i \oslash \mathbf{r}_i$, as follows:

$$[u_{i1}, u_{i2}, \ldots, u_{in}]^T = \left[ \frac{x_{i1}}{r_{i1}}, \frac{x_{i2}}{r_{i2}}, \ldots, \frac{x_{in}}{r_{in}} \right]^T \tag{2}$$

The next day, the rate vector updates to $\mathbf{r}_{i+1} = [r_{(i+1)1}, r_{(i+1)2}, \ldots, r_{(i+1)n}]^T$, and thus, the investment vector of the next day will also update accordingly as follows:

$$\mathbf{x}_{i+1} = \left[ \frac{x_{i1}}{r_{i1}} \times r_{(i+1)1}, \frac{x_{i2}}{r_{i2}} \times r_{(i+1)2}, \ldots, \frac{x_{in}}{r_{in}} \times r_{(i+1)n} \right]^T \tag{3}$$

For this updated investment vector, the total updated investment will be as follows:

$$x_{i+1} = |\mathbf{x}_{i+1}|_1 = \sum_{r=1}^{n} x_{(i+1)j} \tag{4}$$

### 3.1.3. Profit Function

Use these total investments $x_{i+1}$ and $x_i$ of the $i$th and $(i+1)$th days, respectively. The profit $f(\mathbf{x}_i)$ will be their difference, and thus, if the total investment of the next day is increased, then the profit will be positive, otherwise negative. The notation $f(\mathbf{x}_i)$ is used to represent the earned profit depending on the investment of the previous day. The formulation of the profit function $f(\mathbf{x}_i)$ is as follows:

$$f(\mathbf{x}_i) = x_{i+1} - x_i = |\mathbf{x}_{i+1}|_1 - |\mathbf{x}_i|_1 = \sum_{j=1}^{n} x_{(i+1)j} - \sum_{j=1}^{n} x_{ij} = \sum_{j=1}^{n} \frac{x_{ij}}{r_{ij}} \times r_{(i+1)j} - \sum_{j=1}^{n}$$
$$\left( \frac{x_{ij}}{r_{ij}} \times r_{(i+1)j - x_{ij}} \right) = \sum_{j=1}^{n} x_{ij} \left( \frac{r_{(i+1)j}}{r_{ij}} - 1 \right) \tag{5}$$

### 3.1.4. Optimization Problem

At this stage, we have a profit function $f(\mathbf{x}_i)$, which needs to be maximized with respect to the investment. For this purpose, the gradient ascent method was considered, which maximizes the function as follows:

$$\mathbf{x}_{i+1} = \mathbf{x}_i + \eta \nabla f(\mathbf{x}_i) \tag{6}$$

where $\eta$ is the learning rate and $\nabla f(\mathbf{x}_i)$ is the gradient vector of the profit function on the $i$th day. Contrary to the contemporary batch gradient ascent, mini-batch gradient ascent, and stochastic gradient ascent methods, which compute the gradient of the profit function using all dataset points or its sample, the proposed formulation of the profit gradient in Equation (7) depends on instantaneous rate updates only. Depending on the stochastic nature of the forex market rate updates and the employment of only instantaneous rate updates, the proposed methodology is named instantaneous stochastic gradient ascent. Using this profit

gradient, the concrete gradient ascent formulation for our problem to maximize the profit function for the dynamic forex market becomes as stated in Equations (8) and (9):

$$\nabla f(\mathbf{x}_i) = \left[\frac{\partial f}{\partial x_{i1}}, \frac{\partial f}{\partial x_{i2}}, \dots, \frac{\partial f}{\partial x_{in}}\right] = \left[\frac{r_{(i+1)1}}{r_{i1}} - 1, \frac{r_{(i+1)2}}{r_{i2}} - 1, \dots, \frac{r_{(i+1)n}}{r_{in}} - 1\right]^T \tag{7}$$

$$\begin{bmatrix} x_{(i+1)1} \\ x_{(i+1)2} \\ \vdots \\ x_{(i+1)n} \end{bmatrix} = \begin{bmatrix} x_{i1} \\ x_{i2} \\ \vdots \\ x_{in} \end{bmatrix} + \eta \begin{bmatrix} \frac{r_{(i+1)1}}{r_{i1}} - 1 \\ \frac{r_{(i+1)2}}{r_{i2}} - 1 \\ \vdots \\ \frac{r_{(i+1)n}}{r_{in}} - 1 \end{bmatrix} = \begin{bmatrix} x_{i_1} \\ x_{i_2} \\ \vdots \\ x_{i_n} \end{bmatrix} + \eta \begin{bmatrix} \frac{r_{(i+1)1} - r_{i1}}{r_{i1}} \\ \frac{r_{(i+1)2} - r_{i2}}{r_{i2}} \\ \vdots \\ \frac{r_{(i+1)n} - r_{in}}{r_{in}} \end{bmatrix} \tag{8}$$

$$\mathbf{x}_{i+1} = \mathbf{x}_i + \eta\left(\mathbf{r}_{i+1} - \mathbf{r}_i\right) \oslash \mathbf{r}_i \tag{9}$$

This formulation of instantaneous gradient ascent states that the update to the investment vector is directly proportional to the difference of the rate vectors of the consecutive days point-by-point divided by the last day's rate vector. Thereby, as the rate vector is updated, the investment vector is to be updated accordingly for the optimized total investment. The flow diagram of the proposed algorithm is shown in Figure 1. The pseudocode to implement proposed instantaneous stochastic gradient ascent is mentioned in Algorithm 1.

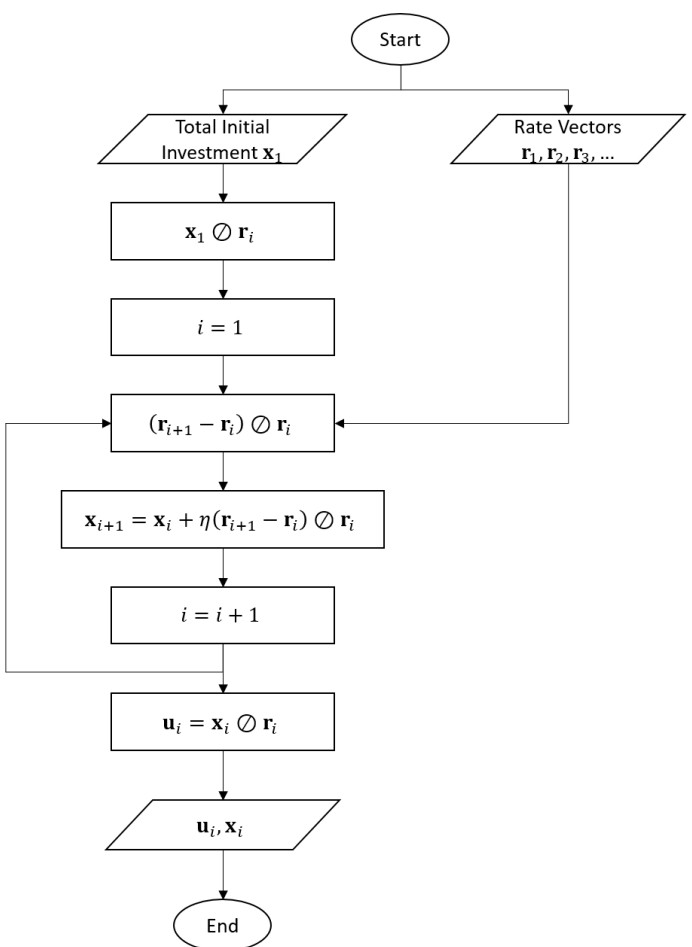

**Figure 1.** Flow diagram to implement the proposed algorithm.

---

**Algorithm 1:** Proposed Instantaneous Stochastic Gradient Ascent Pseudocode

**Inputs** :
Initial investment vector $\mathbf{x}_1$
Rate Vectors $\mathbf{r}_1, \mathbf{r}_2, \mathbf{r}_3, \ldots \mathbf{r}_n$
$\mathbf{u}_1 = \mathbf{x}_1 \oslash \mathbf{r}_1$
**Algorithm** :
$i = 1$
**while** *(rate is being updated)*
$\quad\quad \mathbf{x}_{i+1} = \mathbf{x}_i + \eta (\mathbf{r}_{i+1} - \mathbf{r}_i) \oslash \mathbf{r}_i$
$\quad\quad i = i + 1$
$\quad\quad \mathbf{u}_{i+1} = \mathbf{x}_{i+1} \oslash \mathbf{r}_{i+1}$
**end**
**Output** :
$\mathbf{u}_i, \mathbf{x}_i$

---

## 4. Experiment and Results

The experiment was initially conducted based on an equal amount (RS:200000/-) of investment for each entity, i.e., gold (XD) and four currencies, dollars (USD), Euros (EUR), Dirhams (AED), and Russian rubles (RUB), in the forex market based on the data of the whole year 2020. In our experiments, we selected the learning rate $\eta = 1$. The rates of currencies were obtained from https://www.exchangerates.org.uk/ (access date 1 January 2020), whereas gold rates were obtained from https://www.gold.pk/. To compare with conventional banking sector investments, the KIBOR rates (standard banking investment profit controlled by the state bank of Pakistan) were obtained from https://www.khistocks.com/market-data/kibor-rates.html, as shown in Table 1. Additionally, the updated proposed investment and KIBOR investment are also shown in this table. We note that, although we performed experiments in the whole year 2020 with promising results, considering the limitation of space, the investment updates are reported up to 8th May 2020 only.

Initially, we allocated the investment equally among four different foreign currencies and gold (tola) on the first day. The total amount of investment was around ten-hundred-thousand Pak rupees. We equally distributed the investment of PRK 20,000 for each currency and gold. This is because there was no previous rate of currency and gold. On the next day, new rates of currencies and gold were updated, and the proposed instantaneous gradient ascent method starts to work. The dataset details of various currencies along with the updated proposed investment and updated KIBOR investment are given in Table 1.

**Table 1.** Dataset of various currencies along with updated proposed investment and updated KIBOR investment.

| Date | Dollar (USD) | Euro (EUR) | Dirham (AED) | Russian Ruble (RUB) | Gold (Tola) | Updated Proposed Investment | Updated KIBOR Investment |
|---|---|---|---|---|---|---|---|
| 2 January 20 | 154.88 | 173.64 | 42.17 | 2.51 | 88,300 | 1,000,000.00 | 1,000,273.97 |
| 3 January 20 | 154.90 | 172.61 | 42.17 | 2.49 | 89,650 | 999,268.39 | 1,000,547.95 |
| 6 January 20 | 154.96 | 173.24 | 42.19 | 2.49 | 93,400 | 1,030,970.06 | 1,000,821.92 |
| 7 January 20 | 155.01 | 173.33 | 42.20 | 2.51 | 92,100 | 1,030,970.16 | 1,001,095.89 |
| 8 January 20 | 155.08 | 172.59 | 42.22 | 2.51 | 93,000 | 1,030,970.03 | 1,001,369.86 |
| 9 January 20 | 154.89 | 172.21 | 42.17 | 2.53 | 90,500 | 1,030,970.30 | 1,001,643.84 |
| 10 January 20 | 154.83 | 171.81 | 42.15 | 2.53 | 89,800 | 1,030,970.67 | 1,001,917.81 |
| 13 January 20 | 154.85 | 172.22 | 42.16 | 2.54 | 89,000 | 1,030,970.16 | 1,002,191.78 |

**Table 1.** *Cont.*

| Date | Dollar (USD) | Euro (EUR) | Dirham (AED) | Russian Ruble (RUB) | Gold (Tola) | Updated Proposed Investment | Updated KIBOR Investment |
|---|---|---|---|---|---|---|---|
| 14 January 20 | 154.85 | 172.42 | 42.16 | 2.52 | 89,200 | 1,030,969.90 | 1,002,465.75 |
| 15 January 20 | 154.78 | 172.28 | 42.14 | 2.52 | 89,000 | 1,030,969.99 | 1,002,739.73 |
| 16 January 20 | 154.66 | 172.56 | 42.11 | 2.51 | 89,000 | 1,030,969.71 | 1003,013.70 |
| 17 January 20 | 154.57 | 172.03 | 42.08 | 2.51 | 89,300 | 1,030,970.07 | 1,003,287.67 |
| 20 January 20 | 154.60 | 171.38 | 42.09 | 2.51 | 89,800 | 1,030,970.81 | 1,003,561.64 |
| 21 January 20 | 154.61 | 171.44 | 42.09 | 2.50 | 90,700 | 1,030,970.65 | 1,003,835.62 |
| 22 January 20 | 154.53 | 171.24 | 42.07 | 2.49 | 90,300 | 1,030,970.99 | 1,004,109.59 |
| 23 January 20 | 154.61 | 171.36 | 42.09 | 2.50 | 90,150 | 995,083.48 | 1,004,383.56 |
| 24 January 20 | 154.57 | 170.78 | 42.08 | 2.50 | 90,300 | 995,083.48 | 1,004,657.53 |
| 27 January 20 | 154.57 | 170.35 | 42.08 | 2.47 | 91400 | 995,083.72 | 1,004,931.51 |
| 28 January 20 | 154.57 | 170.29 | 42.08 | 2.45 | 91,400 | 995,083.78 | 1,005,205.48 |
| 29 January 20 | 154.56 | 170.05 | 42.08 | 2.47 | 90,900 | 995,084.04 | 1,005,479.45 |
| 30 January 20 | 154.47 | 170.22 | 42.05 | 2.45 | 91,400 | 995,083.82 | 1,005,753.42 |
| 31 January 20 | 154.49 | 170.28 | 42.06 | 2.44 | 91,500 | 995,083.75 | 1,006,027.40 |
| 3 February 20 | 154.51 | 170.94 | 42.07 | 2.42 | 91,100 | 995,083.04 | 1,006,301.37 |
| 4 February 20 | 154.41 | 170.75 | 42.04 | 2.44 | 90,700 | 995,083.13 | 1,006,575.34 |
| 6 February 20 | 154.49 | 169.91 | 42.06 | 2.45 | 90,750 | 995,083.62 | 1,006,849.32 |
| 7 February 20 | 154.41 | 169.28 | 42.04 | 2.43 | 90,250 | 995,084.55 | 1,007,123.29 |
| 10 February 20 | 154.43 | 169.10 | 42.04 | 2.42 | 90,450 | 995,084.91 | 1,007,397.26 |
| 11 February 20 | 154.42 | 168.55 | 42.04 | 2.41 | 90,700 | 995,086.15 | 1,007,671.23 |
| 12 February 20 | 154.36 | 168.59 | 42.03 | 2.45 | 90,600 | 995086.05 | 1,007,945.21 |
| 13 February 20 | 154.38 | 168.04 | 42.03 | 2.43 | 90,450 | 995,087.55 | 1,008,219.18 |
| 14 February 20 | 154.17 | 167.20 | 41.97 | 2.42 | 90,750 | 995,090.38 | 1,008,493.15 |
| 17 February 20 | 154.28 | 167.38 | 42.00 | 2.44 | 90,800 | 995,089.61 | 1,008,767.12 |
| 18 February 20 | 154.23 | 167.02 | 41.99 | 2.42 | 91,150 | 995,091.03 | 1,009,041.10 |
| 19 February 20 | 154.26 | 166.64 | 42.00 | 2.43 | 92,500 | 995,092.67 | 1,009,315.07 |
| 20 February 20 | 154.24 | 166.62 | 41.99 | 2.42 | 94,300 | 995,092.79 | 1,009,589.04 |
| 21 February 20 | 154.20 | 166.81 | 41.98 | 2.40 | 96,350 | 995,091.91 | 1,009,863.01 |
| 24 February 20 | 154.21 | 167.05 | 41.98 | 2.37 | 96,300 | 1033756.52 | 1,010,136.99 |
| 25 February 20 | 154.26 | 167.49 | 42.00 | 2.36 | 93650 | 1,033,756.51 | 1,010,410.96 |
| 26 February 20 | 154.25 | 168.01 | 41.99 | 2.35 | 94,975 | 1,033,756.73 | 1,010,684.93 |
| 27 February 20 | 154.21 | 168.66 | 41.98 | 2.34 | 95,200 | 1,033,757.35 | 1,010,958.90 |
| 28 February 20 | 154.23 | 170.36 | 41.99 | 2.29 | 92,500 | 1,033,760.07 | 1,011,232.88 |
| 2 March 20 | 154.37 | 170.83 | 42.03 | 2.32 | 92,300 | 1,033,761.59 | 1,011,506.85 |
| 3 March 20 | 154.29 | 171.49 | 42.00 | 2.33 | 92,400 | 1,033,764.05 | 1,011,780.82 |
| 4 March 20 | 154.21 | 172.17 | 41.98 | 2.34 | 94,100 | 1,033,767.05 | 1,012,054.79 |
| 5 March 20 | 154.27 | 171.84 | 42.00 | 2.32 | 94,800 | 1,033,765.40 | 1,012,328.77 |
| 6 March 20 | 154.24 | 173.88 | 41.99 | 2.29 | 94,100 | 1,033,775.15 | 1,012,602.74 |
| 9 March 20 | 156.58 | 178.30 | 42.63 | 2.11 | 94,500 | 1,033,805.31 | 1,012,876.71 |
| 10 March 20 | 157.45 | 178.78 | 42.87 | 2.19 | 97,400 | 1,048,680.99 | 1,013,150.68 |
| 11 March 20 | 158.42 | 179.37 | 43.13 | 2.22 | 96,300 | 1,048,681.07 | 1,013,424.66 |
| 12 March 20 | 159.13 | 179.21 | 43.32 | 2.13 | 94,500 | 1,048,681.72 | 1,013,698.63 |
| 13 March 20 | 158.98 | 177.91 | 43.28 | 2.18 | 93,600 | 1,048,680.74 | 1,013,972.60 |
| 16 March 20 | 158.41 | 177.90 | 43.13 | 2.11 | 89,000 | 1,048,679.80 | 1,014,246.58 |
| 17 March 20 | 158.43 | 175.89 | 43.13 | 2.12 | 89,500 | 1,048,681.40 | 1,014,520.55 |
| 18 March 20 | 158.52 | 173.75 | 43.16 | 2.04 | 89,300 | 1,048,687.47 | 1,014,794.52 |
| 19 March 20 | 158.58 | 172.18 | 43.17 | 2.00 | 87,500 | 1,048,695.33 | 1,015,068.49 |

**Table 1.** *Cont.*

| Date | Dollar (USD) | Euro (EUR) | Dirham (AED) | Russian Ruble (RUB) | Gold (Tola) | Updated Proposed Investment | Updated KIBOR Investment |
|------|------|------|------|------|------|------|------|
| 20 March 20 | 158.68 | 171.19 | 43.20 | 2.04 | 89,900 | 1,048,701.87 | 1,015,342.47 |
| 24 March 20 | 159.01 | 172.57 | 43.29 | 2.02 | 93,200 | 1,048,691.99 | 1,015,616.44 |
| 25 March 20 | 161.61 | 175.23 | 44.00 | 2.09 | 96,100 | 1,048,680.10 | 1,015,890.41 |
| 26 March 20 | 166.13 | 181.75 | 45.23 | 2.11 | 96,600 | 1,048,677.91 | 1,016,164.38 |
| 27 March 20 | 165.54 | 182.72 | 45.07 | 2.13 | 100,100 | 1,048,675.34 | 1,016,438.36 |
| 30 March 20 | 166.14 | 184.24 | 45.23 | 2.08 | 100,100 | 1,048,686.70 | 1,016,712.33 |
| 31 March 20 | 166.70 | 183.14 | 45.38 | 2.13 | 99,500 | 1,048,685.84 | 1,016,986.30 |
| 01 April 20 | 166.83 | 182.74 | 45.42 | 2.11 | 98,500 | 1,048,685.32 | 1,017,260.27 |
| 02 April 20 | 166.93 | 182.27 | 45.45 | 2.14 | 99,300 | 1,048,684.40 | 1,017,534.25 |
| 03 April 20 | 166.77 | 179.99 | 45.40 | 2.16 | 100,600 | 1,048,674.59 | 1,017,808.22 |
| 6 April 20 | 166.99 | 180.27 | 45.46 | 2.19 | 101,300 | 1,048,677.18 | 1,018,082.19 |
| 7 April 20 | 167.90 | 182.36 | 45.71 | 2.22 | 101,700 | 1,048,689.79 | 1,018,356.16 |
| 8 April 20 | 167.76 | 182.06 | 45.67 | 2.21 | 101,500 | 1,048,687.16 | 1,018,630.14 |
| 9 April 20 | 167.19 | 181.71 | 45.52 | 2.24 | 101,600 | 1,048,679.61 | 1,018,904.11 |
| 10 April 20 | 166.79 | 182.43 | 45.41 | 2.26 | 105,200 | 1,048,677.69 | 1,019,178.08 |
| 13 April 20 | 166.83 | 182.38 | 45.42 | 2.27 | 105,100 | 1,048,677.82 | 1,019,452.05 |
| 14 April 20 | 166.95 | 182.38 | 45.45 | 2.27 | 107,200 | 1,108,154.30 | 1,019726.03 |
| 15 April 20 | 166.98 | 182.53 | 45.46 | 2.26 | 107,600 | 1,108,154.30 | 1,020,000.00 |
| 16 April 20 | 166.88 | 181.49 | 45.43 | 2.25 | 107,200 | 1,108,154.13 | 1,020,273.97 |
| 17 April 20 | 163.58 | 177.21 | 44.53 | 2.21 | 106,900 | 1,114,781.47 | 1,020,547.95 |
| 20 April 20 | 163.49 | 178.10 | 44.51 | 2.20 | 103,200 | 1,114,781.40 | 1,020,821.92 |
| 21 April 20 | 161.13 | 174.68 | 43.87 | 2.10 | 103,250 | 1,114,779.05 | 1,021,095.89 |
| 22 April 20 | 160.36 | 174.23 | 43.66 | 2.09 | 101,700 | 1,114,782.31 | 1,021,369.86 |
| 23 April 20 | 159.98 | 172.69 | 43.56 | 2.13 | 101,500 | 1,114,788.39 | 1,021,643.84 |
| 24 April 20 | 160.48 | 172.32 | 43.69 | 2.15 | 102,800 | 1,063,950.55 | 1,021,917.81 |
| 28 April 20 | 161.65 | 175.48 | 44.01 | 2.17 | 102,400 | 1,063,950.60 | 1,022,191.78 |
| 29 April 20 | 161.61 | 175.65 | 44.00 | 2.19 | 102,600 | 1,063,951.09 | 1,022,465.75 |
| 30 April 20 | 160.17 | 174.18 | 43.61 | 2.20 | 103,500 | 1,063,944.41 | 1,022,739.73 |
| 4 May 20 | 159.91 | 174.93 | 43.54 | 2.12 | 102,300 | 1,063,945.89 | 1,023,013.70 |
| 5 May 20 | 159.65 | 174.14 | 43.47 | 2.16 | 102,000 | 1,042,981.22 | 1,023,287.67 |
| 6 May 20 | 160.06 | 172.75 | 43.58 | 2.15 | 101,900 | 1,042,981.23 | 1,023,561.64 |
| 7 May 20 | 160.23 | 173.08 | 43.62 | 2.17 | 102,200 | 1,042,980.84 | 1,023,835.62 |
| 8 May 20 | 159.97 | 173.16 | 43.55 | 2.17 | 103,200 | 1,042,980.60 | 1,024,109.59 |

Table 1 shows that the currency values fluctuate in Pakistani rupees. Figure 2 shows the graphical representation of these fluctuations. It can be seen that fluctuations belong to both upgrading and degrading of the currency values. The last two columns of Table 1 show how the proposed investment strategy and contemporary KIBOR-based banking investment update the invested amount. The comparison shows that the proposed instantaneous-stochastic-gradient-ascent-based investment outperforms all the conventional KIBOR-based investments. However, with its overall performance, there is a constraint associated with the proposed technique, which is observable in the abrupt downgrading in an entity's value.

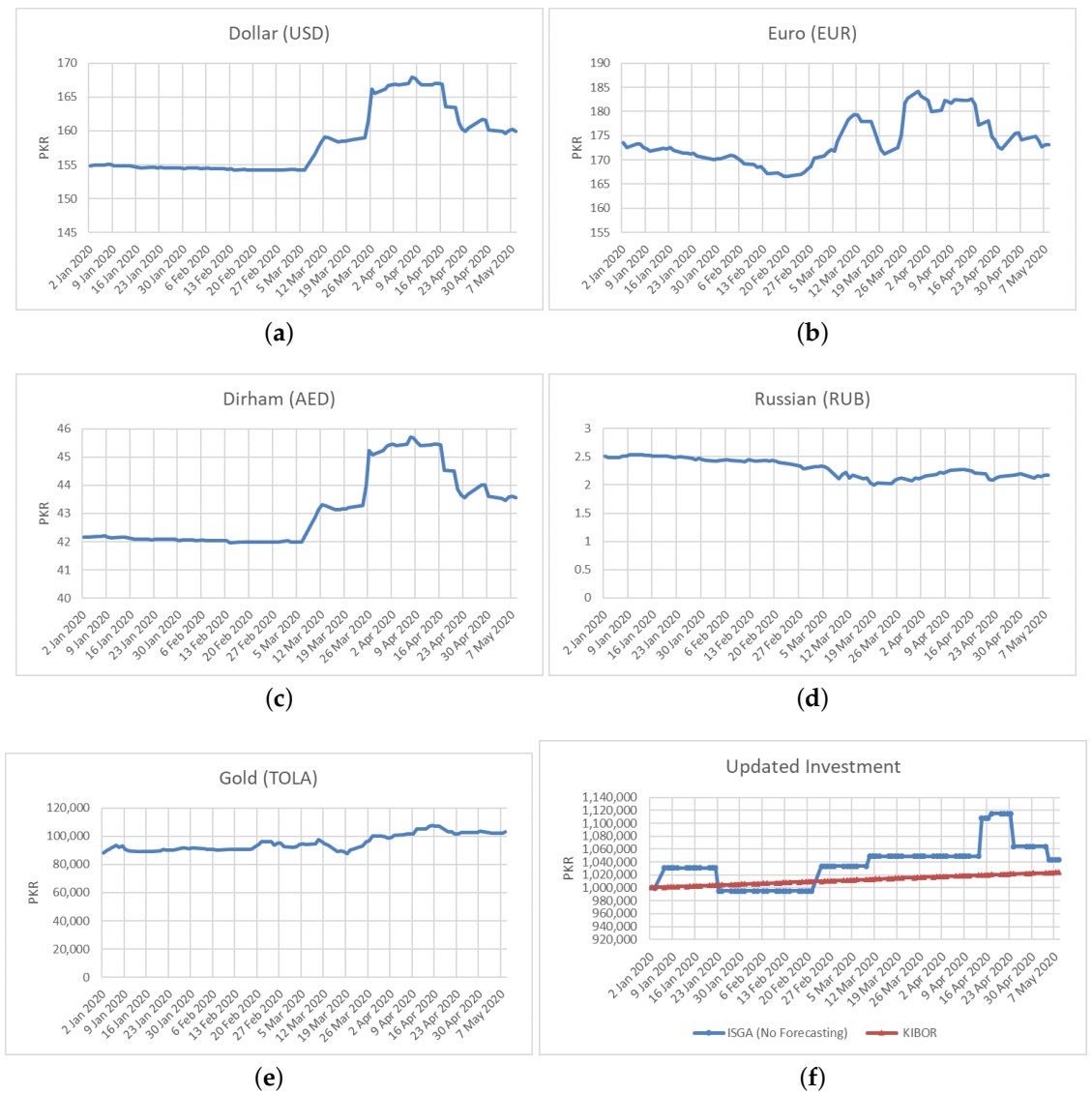

**Figure 2.** Graphical illustration of the currency rates and the comparison of the proposed investment strategy with the KIBOR-based contemporary banking investment. (**a**) Dollar conversion rate to PKR; (**b**) Euro conversion rate to PKR; (**c**) Dirham conversion rate to PKR; (**d**) ruble conversion rate to PKR; (**e**) gold conversion rate to PKR; (**f**) updated investment.

For example, in the month of January 2020, the amount of updated investment using the proposed instantaneous gradient ascent increased up to 3% due to minor changes in the rate of the currencies and gold rate, which are exploited well by the proposed methodology. However, at the end of January, there is a decline in the amount of investment because of the sudden reduction in the rate of gold. This sudden reduction was not exploited by the proposed methodology effectively and, thus, resulted in a loss. This shock was followed by stability and sustainability in the amount of investment till February. The result shows that there was an overall 11% increase in the amount of investment till April 2020 in addition to overcoming the loss of the sudden reduction of the gold rate. This shows the strength of our proposed method in that it works well and acquires profit in the forex market.

It was observed that the proposed investment strategy may have a constrained in-efficacy when there is a sudden reduction in a value of a currency or the gold rate. This does not result in the crashing of forex reserves; instead, this inefficacy is constrained by the sustainable nature of the proposed methodology and, thus, results in minor losses in percentages, which are even recoverable in very few next investment updates in such a

way that the overall invested amount is still outperforming as compared to the contemporary KIBOR investment (see Figure 3). These losses can be reduced by incorporating the appropriate forex forecasting using machine learning.

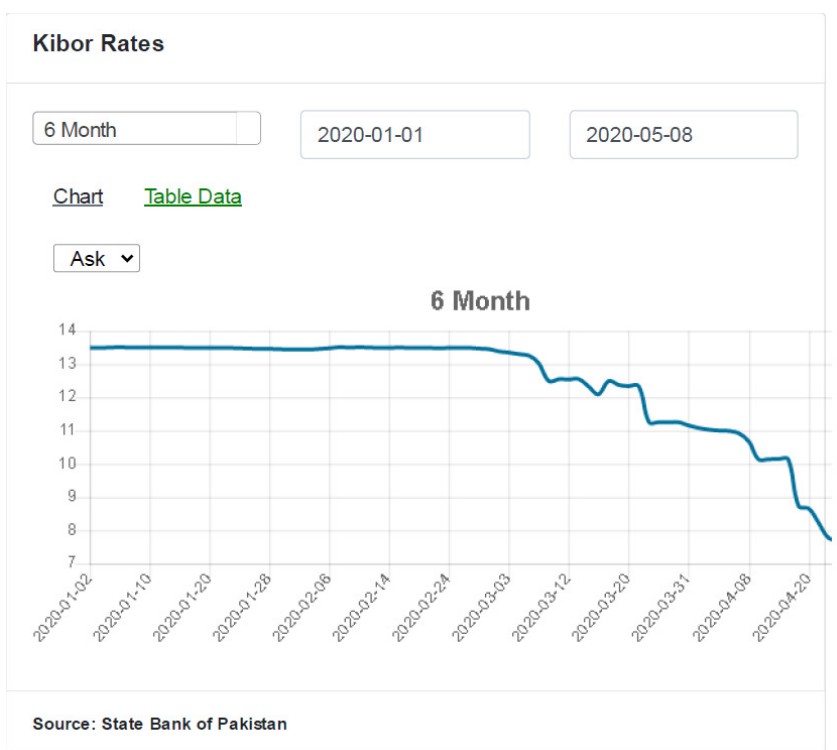

**Figure 3.** KIBOR rate of the year 2020.

In this regard, the forecasting as reported by C. I. Lee et al. [39] and Y. L. Yong et al. [40] is considered as classification. For this, forecasting with a nominal accuracy of 75% was simulated for minimizing this constrained inefficacy in the proposed technique. For this, we considered profit as positive, whereas loss as negative classes. In this regard, if prediction is positive, the forecasting results in investing according to the formulated strategy in the forex market, whereas, if the prediction is negative, the forecasting results in a restraint in investing. Thereby, if the positive prediction is false (false positive), it will result in investment loss, whereas if the negative prediction is false (false negative), it will result in a restraint in investment with missing the chance to obtain the profit. Thus, false positives result in loss, whereas false negatives result in maintaining the investment amount; thus, false positive and false positive have unequal associated costs. In simulating the said nominally, 75% accurate forecasting can perform differently based on the proportion of false positives and false negatives. We considered the worst scenario, i.e., considering the maximum number of false positives as compared to false negatives from 25% false predictions. The performance was then compared with a combined support vector machine and genetic algorithm, random walk, buy and hold, sell and hold, and static genetic-algorithm-based benchmark forex investment strategies for the same time period (2 January 2015 to 2 March 2016) as reported by B. J. Almeida [41].

The results show the proposed instantaneous-stochastic-gradient-ascent-based forex investment when combined with a forecasting technique even with nominally accurate forecasting as shown in Table 2. For comparison, the return on investment (ROI), defined in Equation (10), was considered as employed by B. J. Almeida [41]. We note that, during the pandemic, the host currency (Pakistani rupee) remained relatively stable because of payment relaxation in loans from the World Bank, etc. [42,43], which played a role in the stabilization of the Pakistani economy, as acknowledged by the International Monitory Fund [44]. Since the proposed model exploits currency fluctuations, the performance is

likely to be boosted for those years having more fluctuations as compared to the pandemic year of 2020.

$$ROI(X) = \frac{Return(X) - Investment(X)}{Investment(X)} \tag{10}$$

**Table 2.** Return-on-investment–based comparison of the proposed instantaneous-gradient-ascent-based forex investment with the standard forex investment strategies.

| Investment Strategy | ROI |
|:---:|:---:|
| Random Walk | −22.3 |
| Buy and Hold | −72.2 |
| Sell and Hold | 72.2 |
| Static Genetic Algorithm | 12.5 |
| Average SVM+GA | 43.9 |
| Best SVM+GA | 83.5 |
| Average Proposed ISGA + Forecasting | 80.7 |
| Best Proposed ISGA + Forecasting | 92.9 |

In addition to the time slot, the proposed instantaneous stochastic gradient ascent (ISGA) investment combined with forecasting was also applied to the whole year of 2020, and the results were quite encouraging, as depicted in Figure 4.

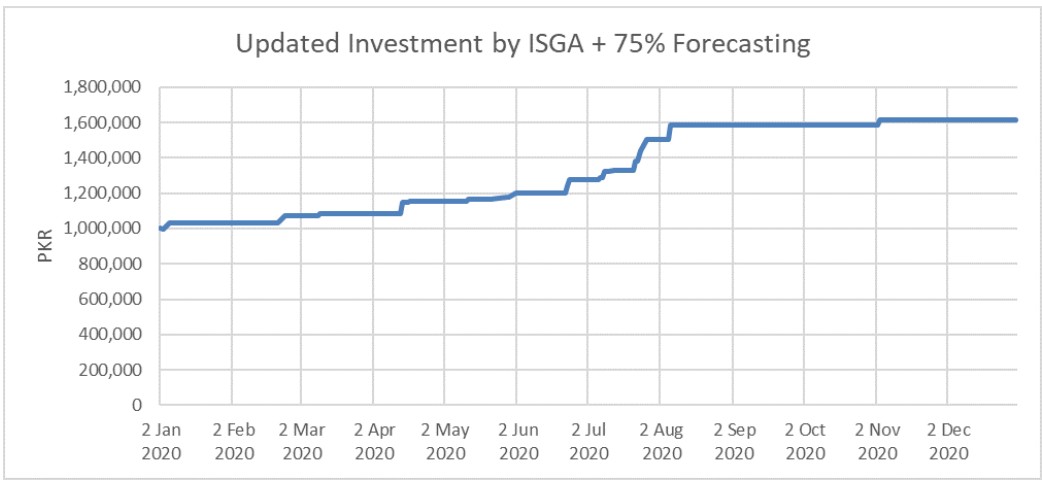

**Figure 4.** Updated investment for the whole year of 2020 using the proposed ISGA on 75% forecasting.

## 5. Conclusions

This research addressed the fluctuating and dynamic time series problem of forex investment in which profitable strategies are critical for sustainable investment. Contrary to the conventional investment strategies, a machine-learning-based automated and adaptive strategy was formulated and implemented in this research. The proposed strategy recommends working in multiple renowned currencies considering the geo-economic situation. The formulation of the proposed methodology was motivated by the stochastic gradient ascent method such that it results in an automated and adaptive investment strategy based on the next time internal forex market situation. In addition to this, the formulation of the proposed instantaneous stochastic gradient ascent method is generic and, thus, can be applied to any base currency, as well as any number of currencies. To validate the formulation of the proposed ISGA method, it was applied to the interbank data of the forex market for the year 2020. The experiments showed that the proposed investment strategy resulted in increased investment sustainability. Unluckily, the proposed investment strategy may have an inefficacy when there is a sudden reduction in the value of a currency or gold rate. Fortunately, this inefficacy is constrained by the sustainable nature of the proposed investment since it considers multiple currencies. In this regard, a larger number

of currencies is more probable in the increased sustainable forex investment. Although this constrained inefficacy is recoverable in the next few investment days, the employment of even a nominal forex forecast increases the investment sustainability effectively.

**Author Contributions:** Conceptualization, I.M. and A.S.; methodology, A.S. and A.I.; validation, R.B; formal analysis, I.M.; resources, A.A.(Abdulkareem Alzahrani); data curation, A.A. (Abdullah Almuhaimeed) and R.B.; visualization, A.A. (Abdullah Almuhaimeed); software, A.A. (Abdulkareem Alzahrani); writing—original draft preparation, A.S.; writing—review and editing, I.M.; supervision, A.A. (Abdullah Almuhaimeed); funding acquisition, A.I. All authors have read and agreed to the published version of the manuscript.

**Funding:** This research received no external funding.

**Institutional Review Board Statement:** Not applicable.

**Informed Consent Statement:** Not applicable.

**Data Availability Statement:** Not applicable.

**Conflicts of Interest:** The authors declare no conflict of interest.

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
