# Peer review of "Forex Investment Optimization Using Instantaneous Stochastic Gradient Ascent—Formulation of an Adaptive Machine Learning Approach"

_sustainability, doi:10.3390/su142215328_

Round 1

Reviewer 1 Report

The subject of the article is timely and interesting for research, but the topic does not fall within the specifics of the journal.

The abstract is not well structured. The Introduction section is quite short and general. Besides the general background, it should be included the literature gap, short information on the method and paper structure. The Literature review is rather superficial. The literature gap is not identified, and the authors' contributions should be highlighted.

The date is not described; the analysed period is not specified. From table 1, figure 3 (which is wrongly labelled figure 2), we deduce that the analysis is limited to the first 4 months of 2020. This period, in addition to being very short, is affected by the beginning of the COVID pandemic, a situation not analysed by the authors.

The results are too briefly discussed, and detailed descriptions and interpretations are required. Also, it is suggested to compare the results of the present research with some similar studies which is done before.

The quality of English should be improved. The paper must be written more clearly so the reader can quickly understand what the author wanted to say.

Author Response

We are thankful to the honorable reviewers for their valuable comments. The authors have taken each comment very seriously and revised the manuscript in light of these valuable comments. The updated portions are highlighted (in yellow) in the revised manuscript. For the sake of minimizing the time of the honorable reviewers, these updates are also included in response to each comment in this file.

Reviewer 2 Report

The proposed procedure is unreasonable in two respects. First, since the future exchange rate r_{i+1} is unknown, this process only applies to backtesting, not to real investments. Second, the loss function f is not well defined. It should be noted that f(x_i) and f(x_j) are different for i≠j.

Author Response

We are thankful to the honorable reviewers for their valuable comments. The authors taken each comment very seriously and revised the manuscript in the light of these valuable comments. The updated portions are highlighted (in yellow) in the revised manuscript. For the sake of minimizing the time of the honorable reviewers, these updates are also included under response to each comment in this file.

Reviewer 3 Report

1. Appreciate that the authors include the background of the countries that have been adopted as cases for the research. why choose them?

2. indicate the contribution of this research paper. 

3. The objective is missing in the introduction. The last sentence in the literature section sounds like the objective. why adopted the method? how about the competing methods? 

4. how was the assumption made from? theory? the practical side? 

5. Why look into this sample period?

6. What was the implication and results of the experiment?

7. The authors should include the forecasting exercise that would add value to the paper.

8. More rigorous analysis should be adopted for the so-called experiments and the sample period should reflect what happens now and in the future.

9. The conclusion would change after considering further estimation and experimentation. with that I would reserve my comments.

Author Response

(The authors gave the same response as above.)

Reviewer 4 Report

Thanks for the opportunity to review this interesting and significant paper.

The section of Literature review is well written. However, the hypotheses miss. They have to be included at the end of literature review part. The section of conclusions focuses on analytical processing and gradually offers a practical interpretation of the results. I appreciate the work on the implications. On the other hand, the absence of limitations and future research ambitions in this area is considered a shortcoming.

Author Response

(The authors gave the same response as above.)

Reviewer 5 Report

The paper is interesting and relevant. However, some aspects deserve to be improved.

In the Introduction, the authors should improve the following aspects:

- better explain the contributions of the article through the results obtained;

- better explain the gaps found in the literature.

In section 2, the authors need to explain what is the forex market; how this market works?

In section 4, the authors must explain the data choice (why January 2020 to April 2020?). Moreover, why are two dates in Table 1 in bold (23/1/2020 and 24/1/2020)?

It also misses the explanation of this Kibor and RS (line 179).

In conclusion, the authors should be clear about the study’s contributions to managers, researchers, investors, and sustainability.

Author Response

(The authors gave the same response as above.)

Round 2

Reviewer 1 Report

Although the answers may have been more complete, I believe that the authors made an effort to improve the article.

Please pay attention to graphical and table representation.

I considered that can be found a method to represent all the obtained results.

Author Response

We would like to thank the Editor for providing us with the opportunity to improve our manuscript. We appreciate the comments and suggestions received from the reviewers, and hereby provide our responses to each comment provided. All changes are highlighted with red color in the revised version.

Reviewer 2 Report

The author addressed all concerns I raised. I recommend that the editor accept the journal.

Author Response

(The authors gave the same response as above.)

Reviewer 3 Report

The earlier comments provided was not been addressed in a proper manner by the authors. Hence, I put back those comments

1. Appreciate that the authors include the background of the countries that have been adopted as cases for the research. why choose them?

2. The authors could use simple sentences of the aim/objective of this research... Additionally, if the last sentence were indeed the objective - automatic and adaptive forex investment strategy by taking the advantage of gradient ascent maximization - which part of the estimation was automatic and adaptive? which model/results that could be the investment strategy for the forex market? specifically which market in this regard?

3. I would suggest the authors present the whole year 2020 results. At least that would be much more updated as compared to the version now. Plus, the authors indicate the results as promising which would be good for policy and proper investment strategy. 

4. Thanks for including Table 2 - however, it is not clear how the estimation of such methods has been carried out. How would that be of comparison to the results obtained in the paper?

5. How the method or analysis consider the pandemic impact?

Author Response

(The authors gave the same response as above.)

Reviewer 4 Report

I have read the entire manuscript carefully and I have not any comments for its authors, because they have addressed all raised issues. 

Author Response

(The authors gave the same response as above.)
